# Establishment of Murine Hybridoma Cells Producing Antibodies against Spike Protein of SARS-CoV-2

**DOI:** 10.3390/ijms21239167

**Published:** 2020-12-01

**Authors:** Nadezhda V. Antipova, Tatyana D. Larionova, Andrei E. Siniavin, Maria A. Nikiforova, Vladimir A. Gushchin, Igor I. Babichenko, Alexey V. Volkov, Michail I. Shakhparonov, Marat S. Pavlyukov

**Affiliations:** 1Department of Functioning of Living Systems, Shemyakin-Ovchinnikov Institute of Bioorganic Chemistry, 117997 Moscow, Russia; nadine.antipova@gmail.com (N.V.A.); iceberg987@yandex.ru (T.D.L.); andreysi93@ya.ru (A.E.S.); shakhparonov@gmail.com (M.I.S.); 2Institute of Medicine, Peoples’ Friendship University of Russia (RUDN University), 17198 Moscow, Russia; babichenko@list.ru; 3Faculty of Biology and Biotechnology, National Research University Higher School of Economics, 101000 Moscow, Russia; 4N.F. Gamaleya National Research Center for Epidemiology and Microbiology, Ministry of Health of the Russian Federation, 123098 Moscow, Russia; marianikiforova@inbox.ru (M.A.N.); wowaniada@gmail.com (V.A.G.); 5E.O. Mukhin Municipal Clinical Hospital, 111399 Moscow, Russia; alex.volkoff@gmail.com

**Keywords:** SARS-CoV-2, Spike RBD, COVID19, hybridoma, monoclonal antibodies

## Abstract

In 2020 the world faced the pandemic of COVID-19 severe acute respiratory syndrome caused by a new type of coronavirus named SARS-CoV-2. To stop the spread of the disease, it is crucial to create molecular tools allowing the investigation, diagnoses and treatment of COVID-19. One of such tools are monoclonal antibodies (mAbs). In this study we describe the development of hybridoma cells that can produce mouse mAbs against receptor binding domain of SARS-CoV-2 spike (S) protein. These mAbs are able to specifically detect native and denatured S proteins in all tested applications, including immunoblotting, enzyme-linked immunosorbent assay, immunofluorescence staining of cells and immunohistochemical staining of paraffin embedded patients’ tissue samples. In addition, we showed that the obtained mAbs can efficiently block SARS-CoV-2 infection in in vitro experiments. Finally, we determined the amino acid sequence of light and heavy chains of the mAbs. This information will allow the use of corresponding peptides to establish genetically engineered therapeutic antibodies. To date multiple mAbs against SARS-CoV-2 proteins have been established, however, bigger sets of various antibodies will allow the detection and neutralization of SARS-CoV-2, even if the virus acquires novel mutations.

## 1. Introduction

At the beginning of 2020 the world faced an outbreak of COVID19 severe acute respiratory syndrome caused by SARS-CoV-2 coronavirus. [1,2]. More than 60 million people were infected during the first 11 months of the pandemic. To slow down the spread of the disease, WHO put significant effort into supporting scientific research and the development of diagnostics, vaccines and medications against COVID19 [3]; additionally, multiple platforms were established to monitor real time distribution of the disease all over the world [4].

Phylogenetic analysis has attributed SARS-CoV-2 to the genus of Betacoronavirus in the Coronaviridae family [5,6]. Despite the name and the genetical similarity, SARS-CoV-2 is not a direct descendant of previously described SARS-CoV virus. Rather it has an independent origin during the evolution [7]. A growing number of full genome sequences of SARS-CoV-2 have revealed multiple mutations and deletions in coding and non-coding regions of the virus [8]. However, SARS-CoV-2 has a relatively low mutation rate due to the high accuracy of the enzymes involved in virus replication [9].

The first step of coronavirus entry into the cell is its interaction with the surface receptors of the host. These receptors include angiotensin-converting enzyme 2 (ACE2) for SARS-CoV and SARS-CoV-2 [10,11,12] and CD26 for MERS-CoV [13]. Spike (S) glycoprotein is responsible for the interaction of SARS-CoV-2 with ACE2. This protein contains a receptor binding domain (RBD) that interacts with the *N*-terminal peptidase domain of ACE2 with Kd of 14.7 nM [14,15,16]. Cryogenic electron microscopy has revealed a molecular structure of ACE2*RBD protein complex [14,15]. Due to its important role and a unique structure, RBD is considered as one of the main targets for the development of neutralizing antibodies against SARS-CoV-2.

In addition, S protein of SARS-CoV-2 contains peptide sequences that may bind to MHC and serve as an effective epitope for the generation of antibodies [17]. It was previously shown that S and N proteins of SARS-CoV induce prominent and prolonged T-cell immune responses [18], which is consistent with recent observation for SARS-CoV-2 [19]. With its currently evolving role, the S protein might serve as a promising antigen for the development of vaccines against SARS-CoV-2 [20]. To this date, more than 20 vaccines based on the S protein are undergoing clinical trials [21].

Besides vaccines, development of diagnostic tools is critically important to stop the spreading of COVID19. The majority of tests for SARS-CoV-2 are being performed using reverse transcription polymerase chain reaction (PCR) on swabs from the nasopharynx or upper respiratory tract. Additionally, computed tomography of the chest is used to confirm the diagnosis, but its results are nonspecific and can often coincide with other diseases, therefore the diagnostic value of this method is limited [22,23]. Serological tests of the immune response of patients are also important, as the presence of specific antibodies allows determination of the prevalence of COVID19 in society and the identification of people who can potentially be immune to the infection. Use of an immunoassay to evaluate the response to the vaccination was also proposed [24,25]. Although the presence of neutralizing antibodies can only be confirmed by specific tests using virus-like particles [26], high titers of IgG detected by immunoassay have been shown to positively correlate with the amount of neutralizing antibodies. To avoid the emergence of virus resistance to the antibodies, the administration of cocktails containing multiple therapeutic antibodies was proposed as a treatment strategy [27,28].

Currently, the antibodies manufacturing industry shares one of the leading places on the market together with the production of vaccines [29]. The number of mAbs based drugs approved for the clinic is growing every year. In addition to human or humanized neutralizing antibodies proposed as a “magic bullet” for the treatment of COVID19 [30], there are a variety of antibodies produced in different species against all proteins of the SARS-CoV-2. These antibodies cannot be directly used to treat patients with COVID19, but they serve as an important tool for both basic scientific research and for the development of vaccines and diagnostic kits for SARS-CoV-2. Unfortunately, none of these antibodies were developed in the Russian Federation. However, during the pandemic it is crucial to have a local source of the antibodies suitable for researches and diagnostics of COVID19.

In the present study we describe the development of hybridoma cells that produce mAbs against RBD of SARS-CoV-2 S protein. We also performed extensive characterization of the obtained mAbs and describe their usage for various applications.

## 2. Results

### 2.1. Mice Immunization with S Protein of SARS-CoV2

Animal immunization is the first step in the development of mouse mAbs. There are multiple proposed protocols to create antibodies against different types of proteins and each of these protocols has its own benefits and drawbacks. Thus, immunization with DNA vectors or cells overexpressing the target protein on their surface allows for the production of antibodies against native protein, which has the correct folding and proper post-translational modifications [31,32]. On the other hand, immunization with synthetic peptides or recombinant protein fragments enables the development of antibodies specific for a certain part of the amino acid sequence without contamination of antibodies raised against oligosaccharides decorating the protein [33]. In the current study, we applied a combined protocol where we first immunized animals with recombinant RBD fragment purified from *E. coli* and then injected mice with cells overexpressing the full-length S protein.

In the first step, we expressed and isolated RBD of S protein from *E. coli* under denaturing conditions (Figure 1A). Despite multiple attempts, we were unable to design a refolding protocol that could allow us to obtain a soluble protein in a buffer suitable for immunization. Therefore, we injected mice with a suspension of insoluble protein in PBS. The schematic representation of immunization schedule is shown in Figure 1B. First, immunization was performed with recombinant protein mixed at a 1:1 ratio with either Freund’s complete adjuvant (FCA) or Freund’s incomplete adjuvant (FIA). Interestingly, FIA allowed to obtain a significantly higher titer of specific antibodies as opposed to FCA (Figure 1C). This result indicates that insoluble RBD is highly immunogenic and does not require any stimulating additives for the development of immune response. Similar data were obtained previously during animal immunization with the surface protein of the Zika virus [33].

Next, immunofluorescence staining (IF) of cells overexpressing the full-length S protein with mice serum demonstrated that antibodies raised against insoluble RBD could detect native S protein (Figure 1D). Therefore, both insoluble RBD and the full-length S protein expressed on the surface of mammalian cells have common epitopes indicating that these proteins could be used for the production and selection of antibodies against SARS-CoV2.

### 2.2. Selection of Hybridoma Cells Secreting mAbs against S Protein of SARS-CoV2

Forty-four days after the first immunization, splenocytes were purified from the mice and subsequently fused with X63 myeloma cells. After the fusion, cells were seeded on 20 96-wells plates and 3 weeks later 120 hybridoma monoclones were obtained. mAbs produced by these clones were tested by enzyme-linked immunosorbent assay (ELISA) against RBD purified from HEK293 cells (Figure 2A). Primary screening with the protein that was purified from the different sources (compared to the protein used for immunization) allowed us to exclude clones that produced antibodies against various contaminants that inevitably present in the samples used for immunization. It is important to note that RBD expressed in HEK293 cells presumably has glycosylation pattern which is similar to that of native S protein and therefore mAbs interacting with RBD from HEK293 most likely could interact with the full-length S.

A total of fourteen hybridomas that demonstrated the strongest immunoreactivity against RBD were chosen for further analysis. Immunoblotting (WB) was performed with the culture medium from our clones to detect the S protein in the lysates of human cells that were transfected with plasmids encoding the full-length S protein or GFP as a control. Representative WBs are shown in Figure 2B. Interestingly, two clones which had highest immunoreactivity in ELISA experiment failed to detect the S protein on WB. Therefore, mAbs produced by these cells are likely nonspecific. Another important observation is that most of the tested mAbs were able to detect endogenous human protein with molecular weight of approximately 60 kDa. It is possible that this protein may have a common epitope with RBD of S protein from SARS-CoV2.

### 2.3. Establishment of Monoclones Producing Antibodies against S Protein of SARS-CoV2

It is well known that hybridoma cells are characterized by the high degree of genetical instability during the first passages. Thus, descendants of a single hybridoma cell will most likely produce different antibodies at early time points after fusion [34]. Therefore, we chose hybridoma #11 which showed highest immunoreactivity against the full-length S protein (Figure 2B) and subcloned these cells to obtain true monoclones. Culture media from 17 subclones of hybridoma #11 were tested by ELISA against RBD purified from *E. coli*, RBD purified from HEK293, and control protein purified from *E. coli* (Figure 3A). Based on our results, we picked 3 subclones for further analysis. Subsequent WB analysis demonstrated that all monoclones produce mAbs with higher specificity compared to the original hybridoma #11 (Figure 3B). It is interesting to note that mAbs from monoclone #11/13 which showed highest signal against RBD purified from *E. coli* but not against RBD purified from HEK293 (Figure 3A) were only able to detect endogenous 60 kDa human protein and showed weak binding to the full-length S protein. Therefore, it is possible that recombinant insoluble RBD has high epitope similarity to the undetermined endogenous protein.

### 2.4. Characterization of mAbs against S Protein of SARS-CoV2

Based on our results (Figure 3) we chose mAbs produced by monoclone #11/9 for further characterization. First, we determined the concentration of mAb #11/9 in cultural medium from hybridoma cells. According to our data the mAb concentration was 19.3 ± 0.7 μg/mL. Next, we used immunofluorescence staining to test if these mAbs were able to bind to native nondenatured S protein. Images on Figure 4A shows that mAbs #11/9 specifically stain human cells that overexpress S protein and have little or no binding to the neighboring nontransfected cells.

It was previously shown that RBD undergoes extensive glycosylation in human cells [35]. Therefore, we aimed to test if glycosylation affects binding of mAbs #11/9 to the S protein. To this end, we treated lysate of cells overexpressing S protein with deglycosylating enzyme PNGase F. Figure 4B demonstrates that deglycosylation decreases molecular weight of S protein by at least 30 kDa, however it has no effect on the intensity of staining with mAbs #11/9. Thus, it is reasonable to conclude that the region of S protein that interacts with mAbs #11/9 is not masked by glycoside groups of the protein.

Finally, we aimed to produce mAbs #11/9 in higher quantities. For this reason, we injected two mice with corresponding hybridoma cells and 12 days later both animals formed ascites. In total 6 mL of ascitic fluid was collected. We used this ascites for serial dilution ELISA with RBD purified from *E. coli*. According to our data, mAbs #11/9 from ascites were able to detect recombinant RBD but not a control protein in as high as 1:1,000,000 dilution (Figure 4C), indicating an efficient binding of the antibodies to the antigen. To further study the reactivity of mAbs #11/9, we used the ascites in 1:10,000 dilution (in this dilution it detects recombinant RBD but not a control protein) and performed ELISA with different amount of RBD that was immobilized in wells of EIA plate. According to our results mAb #11/9 is able to detect as low as 1 ng of recombinant RBD (Figure 4D).

### 2.5. Investigation of Patient Derived Samples Using mAbs #11/9

In clinical practice most samples are prepared as paraffin blocks to facilitate long term storage of the materials. Therefore, we asked if mAbs #11/9 can be used to detect S protein of SARS-CoV2 in paraffin embedded lung tissues by immunohistochemical analysis (IHC-P). According to our data, staining with ascitic fluid formed by monoclone #11/9 taken in 1:1000–1:100,000 dilution allowed to easily distinguish control samples and samples obtained from COVID19 patients (Figure 5A).

mAbs #11/9 were raised against RBD which plays the key role in the virus infection by interacting with the surface protein ACE2 [36]. Therefore, we aimed to test if our antibodies were able to block infection of cells with SARS-CoV2 virus particles. In order to do so, we performed neutralization assay with hCoV-19/Russia/Moscow_PMVL-4 strain of SARS-CoV-2 that was isolated from naso/oropharyngeal swab of COVID19 patient. Using serial dilutions, we demonstrated that mAbs #11/9 can indeed efficiently inhibit SARS-CoV2 infection in in vitro experiments (Figure 5B).

Summing up, we have demonstrated that mAbs #11/9 were able to detect RBD fragment in ELISA experiment; the full-length denatured S protein in WB; the full-length native S protein on a cell surface; the full-length S protein in the lung tissues from the COVID19 patients; and, finally, it can neutralize SARS-CoV2 particles in vitro.

### 2.6. Identification of Amino Acid Sequences of mAbs #11/9

During last few decades genetically modified antibodies (humanized antibodies, single-domain antibody, etc.) were proven to be highly effective for various therapeutic and diagnostic applications [37]. One of the techniques for the production of such antibodies is based on the cloning of the previously identified variable domain sequences from mAbs into the artificially designed vectors encoding constant regions of the antibodies. Therefore, we aimed to determine the amino acid sequence of mAbs #11/9 which it responsible for binding to S protein. For this reason, we amplified cDNA encoding heavy and light chains of mAbs by 5′ SMART RACE method. Using primers specific to k or λ chains we demonstrated that mAbs #11/9 contains only k light chain (Figure 6A). Sequencing of PCR products revealed the nucleotide and amino acid sequence of variable domains of mAbs #11/9 (Figure 6B). Bioinformatic analysis showed that both light and heavy chains do not contain in frame stop codons, have the correct position of the conserved amino acids and that the corresponding nucleotide sequences have not been published previously. Therefore, the obtained sequences most likely represent viable antibodies originating from murine splenocytes but not abortive rearrangement products which theoretically could appear from X63 myeloma cells that were used for fusion.

Finally, we asked if mAbs #11/9 have any similarities to the anti-SARS-CoV2 antibodies which appears naturally in COVID19 patients. To this end, we compared amino acid sequences of complementarity determining region 3 (CDR3) of mAbs #11/9 to the previously published sequences of antibodies expressed by RBD-binding B cells isolated from 60 COVID19 patients [38]. It is important to note that CDR3 sequences were shown to play a major role in the antigen binding and therefore B cells that share an identical CDR3 for both heavy and light chains can be attributed to the same clonotype [38,39]. Comparison of mAbs #11/9 sequences to sequences of 91 different clonotype of RBD binding IgG revealed significant similarity of both heavy and light chains of mAbs #11/9 to the neutralizing antibodies that appears in COVID19 patients after infection (Figure 6C). This result indicates that despite immunization with the artificially synthesized S protein, we were able to obtain antibodies that have similar sequence and most likely bind to the same regions of S protein as the naturally originated SARS-CoV2 neutralizing antibodies.

## 3. Discussion

In this study we describe the development of mAbs against RBD fragment of SARS-CoV2 S protein. Expression and purification of the antigen is often the most challenging step of the mAbs production technique. The optimization of refolding conditions may frequently represent a major problem and time-consuming step in case of proteins expressed in *E. coli*. On the other hand, if protein is expressed in mammalian cells it might be difficult to obtain a sufficient amount of the material for animal injection. Here we utilized immunization of mice with the insoluble protein mixed with FIA and subsequent reimmunization with cells overexpressing the target protein. The main advantage of this protocol is the absence of steps that may require extensive optimization. The only necessary materials are the recombinant protein fragment expressed in *E. coli*, and the plasmid encoding the full-length protein. Both components can be easily obtained using well-established methods. Therefore, we believe that the protocol applied in this study might be useful for the rapid development of mAbs against viral proteins. It became clear during the pandemic of COVID19 that such a necessity may indeed arise in the future.

Although the aim of this study was to create mAbs to S protein, we have also obtained a set of data that might be useful for the development of the vaccines against COVID19. Thus, we demonstrated that two injections of RBD mixed with FIA might be sufficient for the emergence of antibodies that can target the full-length native S protein. On the other hand, it might be worthy to note that most of the mAbs that were established during this study could interact unspecifically with the endogenous protein from human cells. This is consistent with recently published observations, showing that a substantial fraction of naturally occurring SARS-CoV2 neutralizing antibodies show cross-reactivity to self-antigens [40]. This polyreactivity could be explained by the fact that most of the anti-RBD antibodies have a germline or near germline configuration. Thus, it was shown that in COVID19 patients the average somatic hypermutation (SHM) rate of anti-RBD antibodies is relatively low—reaching 2.2–2.8% [40,41]. Consistently, the SHM of mAbs #11/9 was calculated to be 3.4%. In future studies it will be important to determine self-proteins which bind to anti-RBD antibodies and to investigate how these unspecific interactions may affect COVID19 patients.

Another interesting observation could be made during comparison of the WBs in Figure 3B and Figure 4B. The latter one has a much stronger band that corresponds to the S protein. The same cells were used for both WBs, however for Figure 4B we performed cell lysis by boiling in SDS solution, while for the WB in Figure 3B, cells were lysed on ice in an RIPA buffer. Therefore, the discrepancy in WBs might be attributed to the fact that the S protein is tightly anchored to the cell membrane; therefore, harsh conditions are required for its efficient extraction, while a standard lysis protocol leaves most of the S protein in the pellet that contains insoluble cell fragments.

Finally, it is important to note, that for rigorous characterization of the mAbs obtained in this study further experiments are needed. If the mAbs developed here prove to be useful, it will be important to repeat some of the experiments with purified antibodies and to calculate the Kd of antibody-antigen binding. Additionally, it will be interesting to test the ability of these mAbs to inhibit SARS-CoV2 infection using in vivo mice models. However, we believe that the most useful results of the current study are, firstly, the hybridoma cells that are able to produce mAbs against SARS-CoV2 in any quantity needed and, secondly, the amino acid sequence of mAbs against RBD. Knowledge of these sequence might be useful for the development of the cocktails of neutralizing antibodies against COVID19.

In summary, we developed novel mAbs that might serve as an important tool for the scientific research of SARS-CoV2 and that also could help in the establishing of diagnostic and treatment methods for COVID19 patients. The significant advantage of these mAbs is the presence of corresponding hybridoma cells, which could produce these antibodies in nearly any quantity with comparatively little costs.

## 4. Materials and Methods

### 4.1. Plasmid Construction

The DNA fragment encoding RBD of SARS-CoV2 was amplified by PCR using primers S2_for (AAA AGC TAG CAA TGG CAC GAT AAC TGA CGC) and S2_rev (AAT TAA GCT TAA ACA CAT TTG ACC CAG TTG AGT A) from pTwist-EF1a-nCoV-2019-S-2xStrep plasmid kindly provided by Dr. Nevan J. Krogan [42]. Resulting DNA fragment was cloned into NheI/HindIII sites of pET28a+ plasmid (Novagen, Madison, WI, USA) to generate pET28-S plasmid. The absence of unwanted mutations in the inserts and vector-insert boundaries was verified by sequencing.

### 4.2. Recombinant Protein Expression and Purification

To produce RBD fragment of S proteins in bacterial cells, BL21 (DE3) Codone+ RIL *E. coli* cells (Agilent, Santa Clara, CA, USA) were transformed with pET28-S plasmid. Bacteria were incubated at 37 °C in a shaker until OD_600_ reached 0.7. Next, IPTG was added to a final concentration of 1 mM and the bacteria were incubated for an additional 4 h at 37 °C. Then, a quantity of 200 mL media with bacteria were centrifuged for 15 min, 5000× *g* at 4 °C and the pellet was resuspended in 12 mL of lysis buffer B (200 mM NaCl, 100 mM NaH_2_PO_4_, 10 mM Tris-HCl pH 8.0, 8 M Urea, 0.5 mM DTT) and incubated for 1.5 h at room temperature. Next, solution was centrifuged for 15 min, 20,000× *g* at 4 °C and supernatant was incubated with 2 mL of Ni-NTA resin for 1 h under a constant agitation. Suspension was transferred to a column and washed with 20 mL of buffer B and 10 mL of buffer C (same as buffer B but pH 6.3). Bounded proteins were eluted with buffer D (buffer C with 250 mM of imidazole) and dialyzed overnight against PBS with 1 mM DTT. The purity of obtained protein was assessed by electrophoresis and subsequent Coomassie Blue staining. The RBD fragment of the S protein purified from HEK293 cells was kindly provided by Dr. Vassili Lazarev from the Center for Precision Genome Editing and Genetic Technologies for Biomedicine, Federal Research and Clinical Center of Physical-Chemical Medicine of Federal Medical Biological Agency.

### 4.3. Cell Culture

HEK293 (ATCC), Vero E6 (ATCC) and HT1080 (ATCC) cells were grown in Dulbecco’s modified Eagle’s medium (DMEM) supplemented with 10% (*v/v*) fetal bovine serum (FBS), 2 mM L-glutamine, 1 mM Na-pyruvate and penicillin-streptomycin mixture (100 μg/mL). Transfection was performed with Lipofectamine LTX reagent (Thermo Fisher, Waltham, MA, USA) according to the manufacturer’s protocol. Transfected cells were stained or injected into mice 48 h after transfection. X63 myeloma and hybridoma cells were grown in DMEM/F12 medium supplemented with 15% (*v/v*) FBS, GlutaMAX (Thermo Fisher, Waltham, MA, USA), 1 mM Na-pyruvate and penicillin-streptomycin mixture (100 μg/mL). HAT or HT supplements (Sigma-Aldrich, St. Louis, MO, USA) were added at different time points after fusion as described previously [43].

### 4.4. Mice Immunization and Hybridoma Fusion

Five-week-old BALB/c mice were immunized according to the standard protocol [43] with several modifications. First, immunization was performed intraperitoneally with 200 μg of RBD resuspended in 150 μL of PBS and mixed with 150 μL of either Freund’s complete adjuvant (FCA) or Freund’s incomplete adjuvant (FIA) (Thermo Fisher, Waltham, MA, USA). Two weeks later, mice were intraperitoneally injected with 200 μg of RBD resuspended in 150 μL of PBS and mixed with 150 μL FIA. Six days later blood from the tail vein was collected to perform ELISA and IF staining. At days 28, 29 and 30 after first immunization mice were subcutaneously injected with 2∙10^6^ HEK293 cells transfected with pTwist-EF1a-nCoV-2019-S-2xStrep plasmid. At day 40, mice were intraperitoneally injected with 100 μg of RBD resuspended in 100 μL of PBS and 4 days later mice were sacrificed and splenocytes were isolated and subsequently fused with X63 myeloma cells according to the standard protocol [43]. All animal experiments were carried out under an Institutional Animal Care and Use Committee (IACUC) approved protocol #266/2020 in the Shemyakin–Ovchinnikov Institute of Bioorganic Chemistry according to NIH guidelines.

### 4.5. Production of Ascites

Ascites were prepared to obtain large quantities of mAbs [43]. Briefly, mice were intraperitoneally injected with 200 μL of FIA. Five days later, mice were intraperitoneally injected with 2.5∙× 10^6^ of hybridoma cells in 150 μL of PBS. After 12 days, mice were sacrificed and ascitic fluid was harvested from the intraperitoneal cavity and centrifuged at 20,000× *g* at 4 °C for 15 min.

### 4.6. Enzyme-Linked Immunosorbent Assay

Enzyme-linked immunosorbent assay (ELISA) was performed as described previously [44]. Briefly, 0.5 μg of RBD isolated from *E. coli*; RBD isolated from or HEK293 cells or 0.5 μg of a control protein isolated from *E. coli* were immobilized on wells of 96 well EIA plate (Corning, New York, NY, USA). After blocking with 1% BSA in TBST (20 mM Tris pH 7.6, 150 mM NaCl, 0.1% Tween20) wells were incubated with culture medium from hybridoma cells diluted 1:1 in TBST; mouse serum diluted 1:700 in TBST; or ascitic fluid diluted in TBST. After washing with TBST wells were incubated with HRP-conjugated anti-mouse secondary antibodies (Thermo Fisher, Waltham, MA, USA) (1: dilution in TBST) and developed with 1-Step Ultra TMB-ELISA Substrate Solution (Thermo Fisher, Waltham, MA, USA) according to the manufacturer’s protocol.

### 4.7. Antibody Concentration Measurements

Concentration of mAbs in hybridoma culture medium was measured using ELISA as described previously [45].

### 4.8. Immunoblotting

Immunoblotting (WB) was performed as described previously [46]. Briefly, cells were lysed in RIPA buffer (150 mM NaCl, 1% NP40, 0.5% Na-deoxycholate, 0.1% SDS, 50 mM Tris pH 8.0, Protease inhibitor cocktail (Sigma-Aldrich, St. Louis, MO, USA) on ice for 30 min, centrifuged at 18,000× *g*, 4 °C for 10 min and the supernatant was collected. The concentration of total protein in different samples was adjust using BCA assay kit (Thermo Fisher). Same lysates were used to test all culture mediums from different hybridoma cells. Alternatively, cells were lysed by boiling at 100 °C for 10 min in 0.5% SDS and subsequently used for deglycosylation and SDS-PAGE. After electrophoreses proteins were transferred to PVDF membrane and incubated overnight with culture medium from hybridoma cells diluted 1:1 in TBST. After washing membranes were incubated with HRP-conjugated anti-mouse secondary antibodies (Thermo Fisher, Waltham, MA, USA) (1:4000 dilution in TBST with 5% nonfat dry milk). Membranes were developed with SuperSignal West Pico PLUS chemiluminescent substrate (Thermo Fisher, Waltham, MA, USA), analyzed on ImageQuant LAS 500 imager (GE Healthcare, Chicago, IL, USA). Development procedure was performed simultaneously for all membranes represented within the same figure.

### 4.9. Deglycosylation

Deglycosylation of the S protein from the lysate of HEK293 cells transfected with pTwist-EF1a-nCoV-2019-S-2xStrep plasmid was performed as described previously [47] using PNGase F (New England Biolabs, Ipswich, MA, USA).

### 4.10. Immunofluorescence Microscopy

HT1080 cells were plated in wells of Lab-Tek II chamber and co-transfected with pTwist-EF1a-nCoV-2019-S-2xStrep and pTagGFP2-C (Evrogen, Moscow, Russia) plasmids. Two days after transfection cells were washed with phosphate buffered saline (PBS) and fixed with 4% PFA in PBS for 15 min at room temperature. Cells were washed 2 times with PBS and incubated with culture medium from hybridoma cells diluted 1:1 in PBS or with mouse serum diluted 1:700 in PBS. After 5 washes with PBS cells were incubated with AlexaFluor555-conjugated anti-mouse secondary antibodies (Thermo Fisher) (1:500 dilution in PBS) and subsequently stained with DAPI. Images were captured with a DIAPHOT 300 fluorescent microscope (Nikon, Melville, NY, USA).

### 4.11. Immunohistochemistry

With permission from patients’ families, lung samples from two patients who died of COVID19 and two control patients were collected in E.O. Mukhin Municipal Clinical Hospital (Moscow, Russia). The study was approved by the local Medical Ethical Committee (protocol #23/2020) and carried out following the rules of the Declaration of Helsinki of 1975. The diagnoses were confirmed by the history of symptoms, chest computed tomography scans, as well as real-time polymerase chain reaction. Tissue samples were fixed in 10% formalin, embedded into paraffin blocks and sectioned to 5 μm thickness. Immunohistochemistry (IHC-P) was performed as previously described [48]. Briefly, tissues were deparaffinized, and hydrated through an ethanol series. After antigen retrieval in citrate buffer pH 6 slides were processed in Autostainer 360 (Thermo Fisher, Waltham, MA, USA). Staining was visualized with DAB peroxidase substrate.

### 4.12. SARS-CoV-2 Neutralization Assay

The virus neutralization assay was conducted in a BSL-3 facility. hCoV-19/Russia/Moscow_PMVL-4 strain of SARS-CoV-2 that was isolated from naso/oropharyngeal swab of COVID19 patient. Virus was passaged and titrated on Vero E6 cells. Serial five-fold dilutions of ascite formed by monoclone #11/9 were incubated with 100 TCID50 of SARS-CoV-2 for 1 h at 37 °C. The antibody-virus complexes were added to Vero E6 cell-culture monolayer in 96-well plates. The plates were incubated in a CO_2_ incubator at 37 °C for 72 h, after which the cytopathic effect (CPE) was observed microscopically.

### 4.13. Antibody Sequencing

Sequencing of mRNAs encoding heavy and light chains of antibodies was performed using 5′ SMART RACE method as described previously [49] with slight modifications. The first strand of cDNA was synthesized using Mint kit (Evrogen, Moscow, Russia) with primers for k-chain (TTG TCG TTC ACT GCC ATC AAT C), λ-chain (GGG GTA CCA TCT ACC TTC CAG), and heavy-chain (CTG GAC AGG GAT CCA GAG TTC CA) according to the manufacturer’s protocol. Next, cDNA encoding immunoglobulins was amplified by conventional PCR using M1 forward primer (AAG CAG TGG TAT CAA CGC AGA GT) and revers primers specific for k-chain (ACA TTG ATG TCT TTG GGG TAG AAG), λ-chain (ATC GTA CAC ACC AGT GTG GC), and heavy-chain (GGG ATC CAG AGT TCC AGG TC). Obtained DNA was purified and cloned into pKAN-T (Evrogen, Moscow, Russia) vector that was subsequently sequenced. At least 3 different clones were sequenced for each chain of the antibodies. Immunoglobulin sequences were analyzed using IgBLAST (https://www.ncbi.nlm.nih.gov/igblast/), BLASTn (https://blast.ncbi.nlm.nih.gov/Blast.cgi?PROGRAM=blastn&PAGE_TYPE), SignalP-5.0 (http://www.cbs.dtu.dk/services/SignalP/) and Clustal Omega (https://www.ebi.ac.uk/Tools/msa/clustalo/) software.

## Figures and Tables

**Figure 1 ijms-21-09167-f001:**
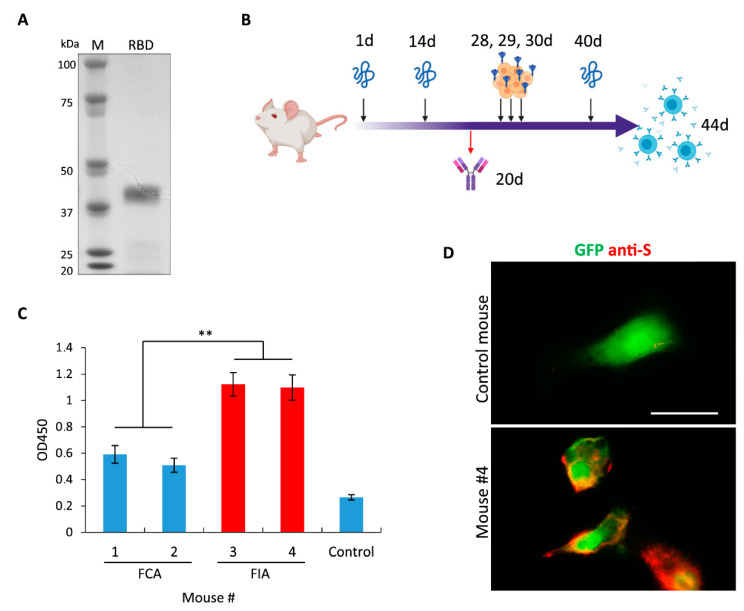
Immune responses of mice injected with SARS-CoV-2 S protein. (**A**) Electrophoresis of recombinant receptor binding domain (RBD) that was purified from *E. coli*. (**B**) Schematic representation of mice immunization workflow. At days 1, 14 and 40 mice were injected with RBD purified from *E. coli*; at day 20 blood was collected to test the presence of Abs against RBD; at days 28, 29 and 30 mice were injected with HEK293 cells overexpressing the full-length S protein; at day 44 mice were sacrificed to isolate splenocytes. (**C**) ELISA showing humoral immune responses of mice immunized with RBD suspension mixed 1:1 with FCA or FIA; non-immunized mouse was used as a control. (**D**) Fluorescence images of cells cotransfected with plasmids encoding GFP (green) and the full-length S protein and subsequently stained with serum obtained from mouse #4 (red). Data are mean ± SD of three replicates. ** *p* < 0.01. Scale bar = 20 μm.

**Figure 2 ijms-21-09167-f002:**
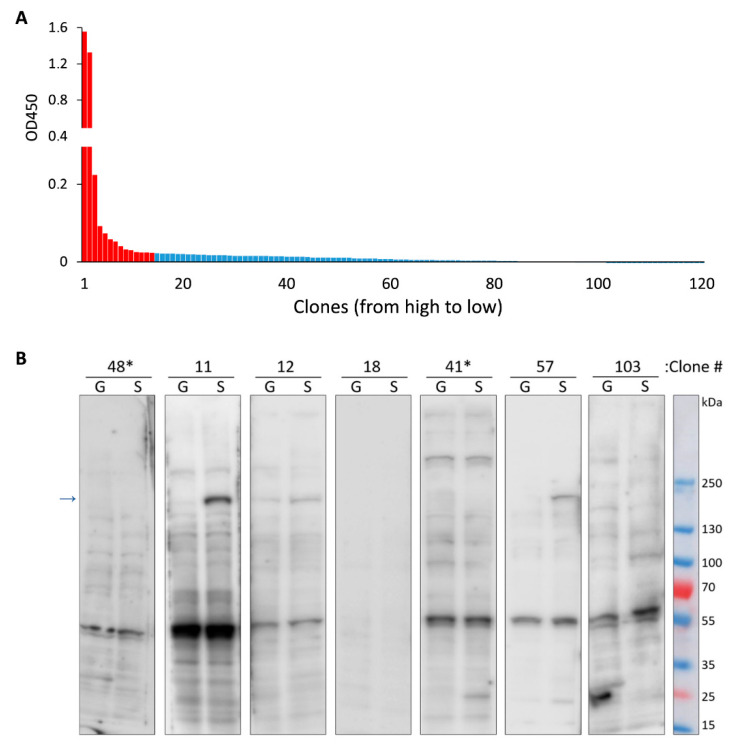
Characterization of Abs secreted by hybridoma monoclones that were obtained after fusion. (**A**) ELISA showing reactivity of Abs from 120 different monoclones to RBD purified from HEK293 cells; red- indicates clones that were used for further analysis. (**B**) Representative immunoblots of cells transfected with plasmid encoding GFP (G) or full-length S protein (S). Culture medium from different monoclones were used to stain the membranes. * indicates clones with the highest reactivity according to ELISA results, arrow indicates the band corresponding to the full-length S-protein.

**Figure 3 ijms-21-09167-f003:**
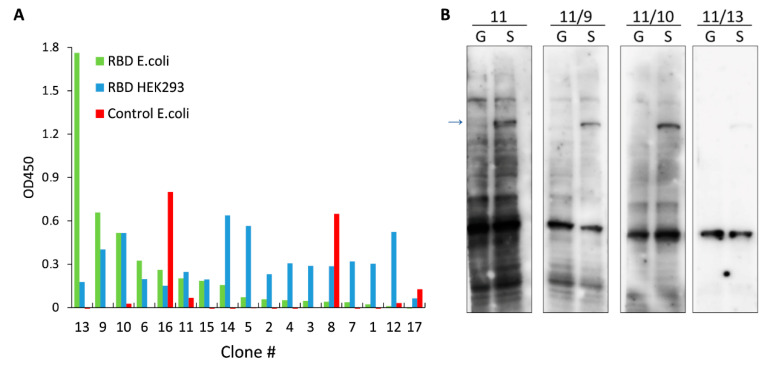
Characterization of mAbs secreted by monoclones that were obtained after subcloning of hybridoma #11. (**A**) ELISA showing reactivity of mAbs from 17 different subclones to RBD purified from *E. coli* (green), RBD purified from HEK293 cells (blue) and to control protein purified from *E. coli* (red). (**B**) Immunoblots of cells transfected with plasmid encoding GFP (G) or the full-length S protein (S). Culture medium from different subclones were used to stain the membranes. Arrow indicates band corresponding to the full-length S-protein.

**Figure 4 ijms-21-09167-f004:**
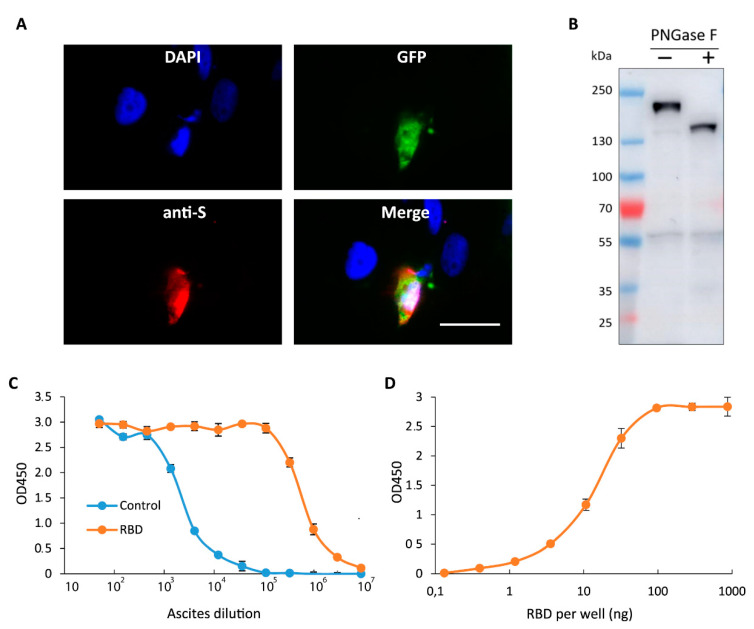
Characterization of mAbs secreted by monoclone 11/9. (**A**) Fluorescence images of cells cotransfected with plasmids encoding GFP (green) and the full-length S protein and subsequently stained with DAPI (blue) and mAbs secreted by monoclone #11/9 (10 μg/mL; red). Scale bar = 20 μm, (**B**) Immunoblot analysis of cells transfected with plasmid encoding the full-length S protein and lysed by boiling in SDS. Lysate was incubated in a presence or absence of PNGase F. Membrane was stained with mAbs #11/9 (10 μg/mL). (**C**) Serial dilution ELISA showing reactivity of ascites formed by monoclone #11/9 against RBD (red) or control protein (blue) purified from *E. coli*. (**D**) ELISA showing reactivity of ascites (1:10,000 dilution) against different amounts of RBD purified from *E. coli* and immobilized in wells of EIA plate. Data are mean ± SD of three replicates.

**Figure 5 ijms-21-09167-f005:**
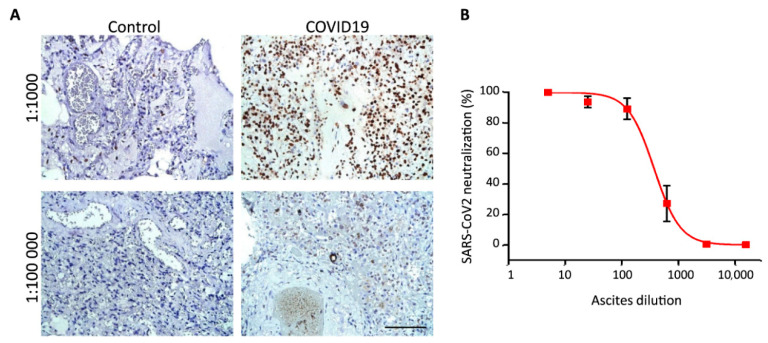
Analysis of clinical samples using mAbs 11/9. (**A**) Immunohistochemical staining of paraffin embedded lung tissue samples from two control and two COVID19 patients. Ascitic fluid formed by monoclone #11/9 was used in 1:1000 and 1:100,000 dilutions. Scale bar = 100 μm. (**B**) Dose-response neutralization curve of SARS-CoV-2 with mAbs #11/9. Data are mean ± SD of five replicates.

**Figure 6 ijms-21-09167-f006:**
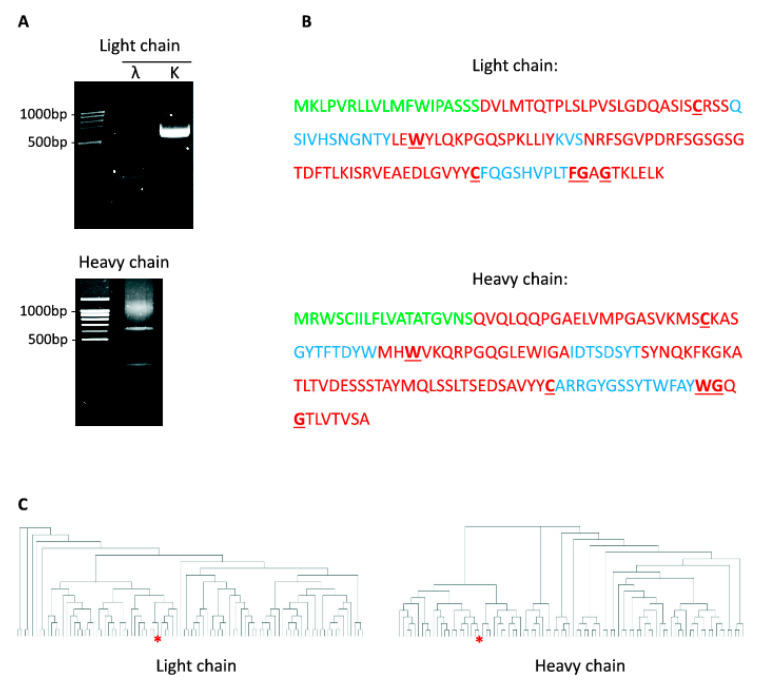
Sequencing of anti-RBD SARS-CoV-2 mAbs. (**A**) PCR amplification of cDNA encoding immunoglobulins’ k-chain, λ-chain, and heavy-chain from monoclone #11/9. (**B**) Amino acid sequence of variable domains from light and heavy chains of 11/9 antibodies’. Different regions of immunoglobulins are highlighted: Ig leader sequence (green); framework regions (red); complementarity determining regions (blue); conserved amino acids (bold, underlined). (**C**) Dendrogram showing clustering of RBD binding antibodies based on the sequences of CDR3 in their light (left) and heavy (right) chains. mAbs #11/9 is indicated with red asterisk.

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
