# Peer review of "Establishment of Murine Hybridoma Cells Producing Antibodies against Spike Protein of SARS-CoV-2"

_ijms, 2020, doi:10.3390/ijms21239167_

Round 1

Reviewer 1 Report

The authors have provided interesting results of good quality in the area of the extremely actual COVID-19 topic.

They have established a hybridoma system to obtain neutralizing antibodies against RBD of S protein of SARS-CoV-2.

The antibodies were purified, concentrated and characterized by a big panel of methods.

The specific novel features of this reserach include (a) an efficient method suggested to get antibodies; (b) the antibodies obtained against the RBD reacted with full length spike protein in several tests; (c) they can neutralize SARS-CoV-2 particles in vitro; (c) good reactivity of the antibodies with lung tissues from the COVID-19 patients was shown; (d) this is the first work regarding anti-SARS-CoV-2 antibodies in Russia. The protocol could be suggested for future medical practice.

There are some suggestions to improve the quality of Methods and Results presentation:

(1) The order of references in the text should be corrected. The last ref. in the Introduction is [30] that is followed by ref. [39,40] in the Results section. The references 31 to 38 are in the Methods section that is in the end of the Ms.

(2) The position of S-protein should be indicated in immuno-blots (Figs 2B, 3B) by arrows or asterisks.

(3) The symbols of Celsius degrees and mathematical degree symbols must be indicated correctly.

E.g. see line 345: "2*106 HEK293 cells" is inappropriate; line 355: "2.5*106 of hybridoma cells" - the same.

Lines 412; 413: indication "at 370C" is inappropriate.

(4) Since the authors suggested presence of cross-reactivity of some of the characterized anti-RBD antibodies with other (host) proteins, they should probably discuss it a bit more wide since this issue is very important for clinical practice (e.g. may be a clue to explain the emergence of auto-immune response at late stages of the disease).

E.g. see https://doi.org/10.1016/j.cell.2020.09.049 for the discussion.

There are also some missprints that should be corrected:

Line 76 "on the marked..." should be replaced with "on the market".

Line 120 "mice #4" should be replaced by "mouse #4".

Author Response

We appreciate the positive comment from the reviewer.

1) We corrected the order and the style of references.

2) We indicated full length S protein by arrows in these pictures.

3) We apologies for these mistakes. We corrected everything accordingly.

4) We thank reviewer for this valuable suggestion. We added the following text to the discussion:

On the other hand, it might be worth to note that most of the mAbs that were established during this study could interact unspecifically with the endogenous protein from human cells. This is consistent with recently published observations that substantial fraction of naturally occurring SARS-CoV2 neutralizing antibodies show crossreactivity to self-antigens [40]. This polyreactivity could be explained by the fact that most of anti-RBD antibodies have germline or near germline configuration. Thus, it was shown that in COVID19 patients average somatic hypermutation (SHM) rate of anti-RBD antibodies is relatively low reaching 2.2% [41] - 2.8% [40]. Consistently SHM of mAbs #11/9 was calculated to be 3.4%. In future studies it will be important to determine self-proteins which binds to anti-RBD antibodies and to investigate how these unspecific interactions may affect COVID19 patients.

5) We’ve proofread the manuscript to correct misprints.

Reviewer 2 Report

Good quality ms, it needs only few modifications. Please check the english, correct some mistakes, cite uptodate references.

Author Response

We appreciate the positive comment from the reviewer. We've proofread our manuscript to improve English and to correct mistakes.